# Recent Advances in Our Understanding of Human Inflammatory Dendritic Cells in Human Immunodeficiency Virus Infection

**DOI:** 10.3390/v17010105

**Published:** 2025-01-14

**Authors:** Freja A. Warner van Dijk, Kirstie M. Bertram, Thomas R. O’Neil, Yuchen Li, Daniel J. Buffa, Andrew N. Harman, Anthony L. Cunningham, Najla Nasr

**Affiliations:** 1Centre for Virus Research, The Westmead Institute for Medical Research, Westmead 2145, Australia; freja.warnervandijk@sydney.edu.au (F.A.W.v.D.); kirstie.bertram@sydney.edu.au (K.M.B.); thomas.oneil@sydney.edu.au (T.R.O.); yuli0321@uni.sydney.edu.au (Y.L.); daniel.buffa@sydney.edu.au (D.J.B.); andrew.harman@sydney.edu.au (A.N.H.); 2Faculty of Medicine and Health, Sydney Infectious Diseases Institute, School of Medical Sciences, The University of Sydney, Sydney 2006, Australia

**Keywords:** plasmacytoid DC (pDC), Axl^+^ Siglec-6^+^ DC (ASDC), DC3, monocyte-derived dendritic cells, epidermal CD11c^+^ dendritic cells, inflammation, HIV

## Abstract

Anogenital inflammation is a critical risk factor for HIV acquisition. The primary preventative HIV intervention, pre-exposure prophylaxis (PrEP), is ineffective in blocking transmission in anogenital inflammation. Pre-existing sexually transmitted diseases (STIs) and anogenital microbiota dysbiosis are the leading causes of inflammation, where inflammation is extensive and often asymptomatic and undiagnosed. Dendritic cells (DCs), as potent antigen-presenting cells, are among the first to capture HIV upon its entry into the mucosa, and they subsequently transport the virus to CD4 T cells, the primary HIV target cells. This increased HIV susceptibility in inflamed tissue likely stems from a disrupted epithelial barrier integrity, phenotypic changes in resident DCs and an influx of inflammatory HIV target cells, including DCs and CD4 T cells. Gaining insight into how HIV interacts with specific inflammatory DC subsets could inform the development of new therapeutic strategies to block HIV transmission. However, little is known about the early stages of HIV capture and transmission in inflammatory environments. Here, we review the currently characterised inflammatory-tissue DCs and their interactions with HIV.

## 1. Introduction

Human immunodeficiency virus (HIV) is predominantly a sexually transmitted virus. Anal intercourse poses the greatest transmission risk in high-income countries [1,2], whilst vaginal intercourse is the primary transmission route in low- and middle-income nations [3,4], with women being more vulnerable than men [5]—particularly young South African women. In both instances, inflammation is highly associated with transmission [6,7,8,9]. This correlation exists regardless of the inflammatory cause, although key drivers include sexual violence trauma [10], sexually transmitted infections (STI) [11,12], bacterial vaginosis (BV) [13,14] and penile anaerobic bacteria, which are common in uncircumcised males [15,16]. Crucially, recent investigations have revealed that the primary HIV preventative treatment, pre-exposure prophylaxis (PrEP), can be ineffective in anogenital inflammation [17,18], which is particularly problematic in sub-Saharan Africa where inflammation is common [19]. This highlights the critical need for better PrEP regimens to block transmission in the context of an inflamed mucosa, yet the early HIV transmission events in an inflammatory milieu remain poorly understood.

Factors that promote inflammation at the site of infection were recently summarised by Caputo et al. [20] and included the ability of different anogenital mucosae (and the pH of the mucus)—whose integrity can be compromised by infections, especially sexually transmitted diseases (including herpes simplex virus), hormonal contraceptives and alterations in microbiome composition—to act as a physical barrier to inflammatory agents. Specifically, microbial vaginal dysbiosis, also referred to as BV, has been identified as a leading cause of the female genital-tract inflammation in sub-Saharan Africa, with the majority of cases being asymptomatic [21,22]. BV-associated inflammation is characterised by a dominant presence of *Gardnerella vaginalis* or *Prevotella* species and correlates with increases in pro-inflammatory cytokine concentrations, including IL-8, IL-1α, IL-1β, and TNF-α [13,23,24]. Contrastingly, those with a ‘healthy’ vaginal microbiota have an abundant *Lactobacillus* species presence, low levels of inflammation and a lower risk of HIV acquisition [25,26,27,28]. While men face a lower risk of heterosexual HIV transmission compared to women, penile microbial dysbiosis can also be a significant contributing factor to HIV acquisition [29]. This risk is particularly high in uncircumcised males as pro-inflammatory anaerobic species, predominantly *Prevotella*, *Dialister* and *Peptostreptococcus*, colonise the inner foreskin [15,16]. An inflammatory rectal microbiome signature also plays a major role in HIV acquisition, though its impact is more complex due to the interplay between external environmental factors and the gut microbiome [30,31].

Inflammation rapidly transforms the cellular landscape of a tissue, resulting in an influx of immune cells, changes to resident cells and a complex network of signals from pro-inflammatory cytokines and chemokines. This dynamic and intricate nature of inflammation poses significant challenges for research, which in part accounts for the incomplete understanding of the relationship between anogenital inflammation and HIV acquisition. The proposed factors contributing to HIV susceptibility in an inflammatory anogenital environment include breached barrier integrity, phenotypic changes in tissue-resident cells and an influx of recruited immune cells. Individuals with an inflammatory profile have an impaired anogenital epithelial barrier, owing to the production of destructive inflammatory proteases [32,33,34], exposing the underlying connective tissue, which hosts an array of HIV target cells. Additionally, in an inflamed environment, tissue-resident HIV target cells, including dendritic cells (DCs), macrophages and CD4 T cells, undergo phenotypical and functional changes that make them more susceptible to HIV infection. Under a pro-inflammatory cytokine stimulation, DCs become activated and mature, increasing their likely uptake of HIV and transfer to CD4 T cells [35,36,37,38]. Lastly, vulnerability to HIV infection may be due to the recruitment of HIV target cells in response to inflammatory signals. CCR5^+^ cells are specifically enlisted by the chemo-attractant chemokines CCL3, CCL4 and CCL5 released by the infiltrating inflammatory plasmacytoid DC (pDC) [39,40]. As CCR5 is a key HIV entry receptor, specific HIV target cells, including CD4 T cells, monocyte-derived macrophages (MDMs) and DCs, are increased in inflammation. Very little work has been carried out in the genital tract to define dendritic cell responses to viruses or other microbial products. Therefore, future investigations into which dendritic cells are responding to sexually transmitted viruses (such as HSV-2 or pathogenic species present in dysbiotic genital tissue, such as *Gardnerella vaginalis*, or *Prevotella*) may identify therapeutic targets to interrupt the ongoing recruitment of immune cells in response to pathogenic microbiomes. This review will provide a comprehensive exploration of the currently defined inflammatory tissue DCs and their known interactions with HIV.

## 2. Human Anogenital Tissue

Given that sexual transmission occurs within genital and anorectal (anogenital) tissues, understanding the anatomy of these tissues, which HIV penetrates, is important (Figure 1). Anogenital tissue comprises three distinct tissue types: skin, type I mucosa and type II mucosa, which all differ in their structural organisation and composition of immune cells and hence vary in their susceptibility to HIV infection. Skin comprises two distinct layers—an outer epidermis and an underlying dermis—and covers the most outer anatomical structures of the body, including the labia major, labia minora, outer foreskin, glans penis, and anal verge. The epidermis forms an impermeable physical barrier to invading pathogens, consisting of a thick stratified squamous epithelium and a surface layer of dead cornified cells—the stratum corneum. Skin is the most robust barrier against HIV transmission. Type II mucosa includes the vagina, ectocervix, inner foreskin, penile fossa navicularis and anal canal. It consists of a stratified squamous epithelial and underlying connective tissue layer called the lamina propria. However, in contrast to skin, it has no (or a very thin) stratum corneum and is therefore more susceptible to HIV infection. The endocervix, urethra, rectum, and colon are all covered by type I mucosa, which comprises only a single layer of columnar epithelium designed for adsorption and the underlying lamina propria. This is the most fragile tissue type and is most vulnerable to HIV penetration and infection.

## 3. Dendritic Cells in Human Tissue

DCs are potent antigen presenting cells (APCs) and form part of the mononuclear phagocyte (MNP) family, which also consists of Langerhans cells (LCs), macrophages and monocytes. They are innate immune cells that form the first line of defence by sampling pathogens via their surface receptors to subsequently trigger an adaptive immune response. Like LCs and macrophages, DCs are also HIV target cells as they express the key HIV entry receptors CD4 and CCR5. They reside in anogenital tissues, and as part of their APC function, they capture HIV at the mucosal entry point and transport the virus to the primary HIV target cell, the CD4 T cell, either within the tissue or nearby lymph nodes.

The development of dendritic cells and macrophages is summarised in Figure 2. Traditionally, DCs have been classified into two lineages: conventional DCs and inflammatory plasmacytoid DCs (pDCs). DCs are further divided into DC1, DC2 and DC3, with distinct ontology and functions. DC1s uniquely have a high capacity to activate CD8 T cells [41]. However, they are resistant to HIV infection [38,42]. DC2s are the most abundant DC population. They are potent naïve T-cell stimulators and highly efficient at transmitting HIV to CD4 T cells [38,43]. DC3s, a recently defined subset, have similar T-cell-activation capacity to DC2s but are ontologically distinct [44,45,46,47]. Recent investigations have revealed DC3s to be highly implicated in inflammation. LCs are restricted to the epidermis and have similar functions to DC2s, although in mucosal tissues they have weaker APC capabilities [43]. Macrophages are either tissue-resident autofluorescent macrophages [48] or monocyte-derived macrophages (MDMs). They phagocytose foreign pathogens and maintain tissue homeostasis. However, they have only a secondary antigen-presenting function. Monocytes infiltrate tissue from the blood where they can differentiate into monocyte-derived dendritic cells (MDDCs) or MDMs [49] based on the cytokine and chemokine signals.

Investigations to date characterising MNPs in human tissue have primarily been performed in steady-state conditions. Since anogenital inflammation is now recognised as a key factor in HIV transmission, with epidermal LCs and DCs being the first target cells encountered, investigating and understanding the inflammatory landscape of DCs and their interactions with HIV is urgently needed to reduce the risk of HIV acquisition in the anogenital mucosa. To date, the recognised inflammatory DCs include plasmacytoid DCs (pDCs), Axl^+^ Siglec-6^+^ DCs (ASDCs), DC3, CD14^+^ and MDDCs (Figure 3).

## 4. Dendritic Cell—T-Cell Transmission Mechanisms

DCs can transfer HIV to CD4 T cells via two distinct mechanisms: first-phase transfer, also referred to as *trans*-infection, and second-phase transfer, or *cis*-infection [37,50,51,52]. First-phase transfer is dependent on lectin receptor-mediated uptake into vacuolar virus-containing compartments (VCCs) and occurs independently of DC infection [53]. These VCCs are continuous with the plasma membrane, connected to the external environment via a narrow canal, at neutral pH and poorly accessed by antibodies. If the virus encounters CD4 T cells within a few hours, a viral synapse forms with contacting CD4 T cells, allowing transfer to and the infection of the T cells. However, within 12–24 h, the majority of the virus is degraded, but it has not yet been shown how or if the virus leaves these neutral pH compartments for end lysosomal degradation [37]. Currently, there are five known HIV-binding lectin receptors: Langerin/CD207/CLEC4K, Mannose Receptor (MR)/CD206/CLEC13D, DC-SIGN/CD209/CLEC4L, Siglec-1/CD169 and DCIR/CD367/CLEC4A. Langerin has been shown to mediate HIV uptake by epidermal Langerhans cells and transfer to CD4 T cells [54], although there is some viral degradation within the Birbeck granules [55]. Sub-epidermal langerin^+^ DC2s are also very efficient at HIV uptake but they express langerin at 10-fold lower levels than LCs, and it is not known if this langerin expression mediates HIV uptake by these cells. MR is expressed by most tissue DCs as well as macrophages and can mediate HIV uptake [37,56]. However, binding is weak and leads to viral destruction by lysosomes rather than uptake into caves [37]. DC-SIGN is highly efficient at HIV binding and uptake [57], and although this was originally thought to be a DC receptor, recent evidence demonstrates that DC-SIGN is expressed by macrophages [38]. Siglec-1 is expressed by most MNPs and inducible by type-I interferon (IFN-I) production [58]. Recently, it was induced at the gene level in macrophages by type III interferon (l3), although this remains to be validated at the at the protein level [59]. It has been shown to directly mediate HIV uptake into intracellular caves in in vitro-generated MDDCs, but only when in a mature state [60,61,62,63]. Siglec-1 can also act to concentrate HIV and transfer it to the CD4/CCR5 entry receptors to facilitate productive infection [37,64]. However, like DC-SIGN, Siglec-1 is now known to be expressed predominantly by macrophages, with the notable exception of a subset of inflammatory ASDCs, as discussed below. CLEC4A has also only been explored in in vitro-generated MDDCs, on which it is capable of HIV uptake and transfer [65,66], albeit not as efficiently as DC-SIGN [67]. CLEC4A is expressed highly by all tissue MNPs; however, it has not yet been shown capable of binding or transmitting HIV in tissue. The varying expression of lectin receptors across MNP subsets not only implicates their roles in first-phase transfer but further serves an additional way to classify MNP subsets.

Second-phase transfer is dependent on productive HIV infection via the entry receptors CD4 and CCR5, which mediate the fusion of the virus envelope with the DC plasma membrane and the release of the viral genome into the cytosol. It occurs 72 h after initial infection as de novo-produced virions bud from the DC plasma membrane and infect CD4 T cells as they interact with DCs via a virological synapse [68,69,70].

## 5. Plasmacytoid Dendritic Cells

### 5.1. Origins and Discovery

pDCs were first described in the 1950s in human lymph nodes and were initially thought to be T cells based on their surface protein expression profile [71]. Several decades later, upon being cultured, they were found to develop DC-like morphology and express high levels of MHC proteins [72,73], hence gaining their classification as a DC (although this ‘DC’ classification remains a topic of debate [74,75,76]). Traditionally defined as CD123^+^ CD11c^−^, pDCs are the primary IFN-I-producing cells in response to viral infection [77,78] and crucial mediators of the inflammatory response [79]. pDCs are derived directly from the common DC progenitor (cDP), which arises from IRF8^high^ granulocyte monocyte DC progenitors (GMPDs) like DC1 and DC2 [47]. Similar to other DCs derived from a myeloid lineage, pDC differentiation can be driven by FLT3/CD135, c-Kit/CD117 and M-CSF/CD115 [80], but uniquely, pDCs can also be derived directly from the lymphoid-primed multipotent progenitor regulated by the transcriptional factor TCF4/E2-2 [81,82].

### 5.2. Inflammation and Immunological Functions

pDCs’ high expression of toll-like receptor (TLR) 7, TLR9 and IRF7 enables their potent production of IFN-I [44,83]. The production of IFN-I (predominantly IFNα but also IFNβ) is their most fundamental immunological role, which forms one of the first and most important innate immune responses in restricting viral transmission. pDCs are particularly well-recognised in their initial responses to HIV infection [40,84,85,86]. They migrate from blood to sites of infection within 1–2 days of exposure, where they mount a strong inflammatory response through the production of pro-inflammatory cytokines, including TNFα and IL-6 [84,87,88], and CD4 T-cell-recruiting chemokines CCL3-5 and CXCL8 [40,84,87]. Therefore, the role of pDCs during HIV infection is controversial as it is not clear whether these cells exacerbate or limit initial infection. Since the discovery of ASDCs, which were found to be contained within the CD123^+^ pDC population [44], pDCs have recently been redefined to have a limited ability in inducing T-cell activation, proliferation and polarisation [87,89].

After excluding ASDCs, several studies have identified and characterised three subpopulations within the pDC compartment, all with distinct phenotypes and functionalities that arise upon activation with different stimuli [90,91,92]. In 2018, Alculumbre et al. [90] identified three subsets—P1, P2 and P3—of activated Axl^−^ pDCs upon stimulation with TLR7 ligands (influenza and R848), TLR9 ligands (CpG-A, CpG-B and CpG-C), bacteria (S. aureus) or cytokines (IL-3 and GM-CSF). P1-pDCs (PD-L1^+^ CD80^−^) maintained a plasmacytoid morphology and high IFN-I production, whereas P3-pDCs (PD-L1^−^ CD80^+^) developed a dendritic morphology, had elevated CCR7 expression and were capable of T-cell activation and Th2 polarisation, but they were unable to produce IFN-I. P2-pDCs (PD-L1^−^ CD80^+^) displayed a functional continuum of both P1 and P3 and likely are an intermediate cell. TLR7 ligands and CpG-Ca significantly induced all three pDC populations, CpG-A and *S. aureus* primarily induced P1-pDCs, whereas CpG-B, GM-CSF and IL-3 preferentially induced P3-pDCs. Furthermore, they showed that only P1-pDCs were detected in samples (blood or skin) from patients with psoriasis or lupus. Alculumbre et al. also found that the specialisation of these pDCs into P1, P2 or P3 was independent of a pre-existing heterogeneity, a pre-commitment of pDC precursors or IFN feedback but rather a result of cell-to-cell communication from autocrine or paracrine loops of cytokine secretion. Onodi et al. [91] later showed that human pDCs efficiently diversified into functional P1-, P2-and P3-pDC effector subsets in response to SARS-CoV-2, while Cuevas et al. [92] demonstrated that P3-pDCs accumulated within tumours of melanoma patients and negatively correlated with clinical outcomes. These new discoveries of pDC plasticity and complexity prompt the need to further investigate pDCs’ specific functional roles in immunity and inflammation.

### 5.3. HIV Interactions

pDCs’ role in HIV infection serves predominantly as a protective function. Upon infection, pDCs become activated by HIV RNA through the TLR7 signalling pathway which triggers a rapid IFN-I response [84]. This defensive viral mechanism was first evidenced through SIV infection in rhesus macaque models [93]. Upon blocking the IFN-I receptor preceding an acute rectal SIV infection, there was a rapid depletion of CD4 T cells and expansion of the viral reservoir, resulting in an accelerated onset of AIDS. Indeed, several studies reported IFN resistance to be a major contributor to HIV transmission [94,95], emphasising the importance of pDCs’ antiviral role.

It was previously believed that pDCs were capable of being productively infected by HIV [96,97]; however, this has since been attributed to the contaminating CD123^+^ ASDC population [64,87]. In fact, the pro-inflammatory chemokines CCL3-5 secreted by pDCs in response to HIV simultaneously recruit CD4 T cells and block HIV infection by binding to the co-receptor CCR5 [98]. This blockage of the HIV entry receptor and high IFN production both work to inhibit the productive infection of pDCs [84,86]. One study observed pDCs to be capable of HIV transfer to CD4 T cells at a very limited capacity in first-phase transfer only [87]. This may be explained by their expression of CLEC4A, a C-type lectin that has been shown to bind HIV [65]. The diverse roles of pDCs in HIV interactions also raise questions about the functions of activated P1-, P2-, and P3-pDC subsets [90]. P3-pDCs morphologically resemble DCs and, combined with their high expression of co-stimulatory molecules, migratory capacity and ability to activate and proliferate CD4 T cells, it could be hypothesised that P3-pDCs are able to perform HIV transfer in the first phase. P1-pDCs were the only subset that produced IFN; hence, the rapid IFN response triggered by HIV exposure could be attributed solely to P1-pDCs. Further investigation of these pDC subpopulations’ interaction with HIV is needed to fully understand their specific and individual roles in HIV transmission.

The activation of TLR7 to induce HIV expression in latently infected cells through the release of IFNα from pDCs [99] has been explored. However, this approach is limited by the following: (1) TLR7 polymorphism, which affects the ability of pDCs to activate and secrete IFN [100]. (2) A decline in pDC numbers, which occurs in most people living with HIV (PLH). Upon the initiation of antiretroviral therapy (ART), their numbers can increase but are not fully restored. (3) pDCs are dysfunctional upon ART interruption, thus interfering with TLR signalling. (4) pDCs predominately produce IFNα2, which is less potent than IFNα8/14 [101]. (5) Only long-term non-progressors and elite controllers of PLH have preserved pDCs counts and functionality [102]. Therefore, targeting pDCs for an HIV cure has many limitations, but, potentially, a direct IFNα8/14 treatment may be a better approach to reactivate latent HIV [86], especially as IFNα8/14 can reshape the human immune system by activating macrophages, natural killer cells, DCs, and T cells and secreting a variety of cytokines that modulate B- and T-cell activation [103].

## 6. ASDCs

### 6.1. Origins and Discovery

ASDCs are a newly defined DC population initially identified in human blood, owing to advances in high parameter single-cell technologies; they are distinguished by their unique expression of Axl, Siglec-6 and CD5 [44,89,104]. ASDCs can be further divided into two discrete subsets by the expression of CD11c or CD123, with each subset having distinctive phenotypical and functional profiles [44,87]. The CD123^+^ and CD11c^+^ ASDC populations were identified by Villani et al. [44] to express AXL, Siglec-6, CD2, CX3CR1, CD33, and CD5, similar to the phenotype described for pre-DC1 and pre-DC2 (CD33^+^ CD45RA^+^ CD123^+^ CX3CR1^+^ CD2^+^ CD5^+^ Siglec-6^+^) identified by See et al. [89]. Therefore, CD123^+^CD11c- ASDCs are pre-DC1s, while CD123^−^ CD11c^+^ ASDCs are pre-DC2s, with CD11c^+^ ASDCs being a different population from DC2 [87]. ASDCs infiltrate peripheral tissues during inflammation, while cDC2s are present in both healthy [38] and inflamed tissues [105].

The discovery of these new DCs has prompted a rapid reclassification of pDCs’ immune function capabilities as many of their previously described roles, including antigen presentation, T-cell activation and also HIV infection [96,97], have now been attributed to contaminating ASDCs [44,64,87,89].

The developmental origins of ASDCs remain contentious; some studies report ASDCs to be an intermediate cell state—a precursor to DC1 or DC2 [89], whilst others detail a distinct and functional DC, reminiscent of a bona fide population [44]. Like pDCs, DC1 and DC2, ASDCs arise from the cDP. See et al. [89] further contends that ASDCs are circulating DC progenitors, transient pre-DC1s or pre-DC2s that commit to DC1/2 differentiation. In contrast, Villani et al. [44] found ASDCs to be a well-defined cell subset, with morphological similarities to DC2s, high T-cell proliferative capacity and distinct functionalities, including a lack of IFN production and secretion of IL-12p70 and IL-8. In support of this, Alcántara-Hernández and colleagues [104] further demonstrated that when cultured, ASDCs only transitioned towards a DC2 phenotype and not DC1. They also observed that ASDC distribution in inflamed skin and lymphoid organs mirrored pDCs and not DCs, challenging the idea that ASDCs are DC precursors. Importantly, Warner van Dijk et al. showed that ASDCs derived from blood and tissues are a separate population to DC2s but are mature in phenotypes like DC2s [87]. Further studies are needed to investigate the developmental origins and differentiation potential of these cells before firm conclusions are made [73].

### 6.2. Inflammation and Immunological Functions

In peripheral tissue and lymphoid organs, ASDCs, like pDCs, are inflammatory cells that infiltrate tissue from the blood in response to stimuli. A recent investigation compared an inflamed rectum from an ulcerative colitis patient to a non-inflamed rectum and found ASDCs to be exclusively present in the inflamed sample [87]. This study further identified ASDCs in the inflamed colon of diverticulitis and colorectal cancer patients and on psoriasis-diseased skin, inflamed mesenteric lymph nodes and inflamed anogenital tissues, including those from the labia and foreskin. ASDCs have also been identified in a range of other inflammatory conditions and locations, including inflamed cerebrospinal fluid (CSF) from multiple sclerosis (MS) patients [106], skin from blisters and wounds [107], inflamed bronchoalveolar lavage [108], tonsils [44,104] and spleens [89,104] (Table 1).

Many ASDC functional investigations were limited by two main factors: (1) since ASDCs are infiltrating inflammatory cells, the tissue-derived populations were too small for functional studies [87,108], so blood-derived ASDCs were used instead, and (2) few studies separated ASDCs into their discrete CD11c^+^ and CD123^+^ subsets [44,87,104,108,109], meaning that most functional data were derived from a combined ASDC population or only the CD123^+^ subset. Nonetheless, in blood, ASDCs have higher T-cell-stimulating capacity and a more mature phenotype compared to pDCs, similar to DC2s [44,87,89,104,109]. Warner van Dijk et al. [87] further showed that CD11c^+^ ASDCs expressed higher levels of the costimulatory molecules CD86, ICAM-1 and HLA-DR and hence were more potent inducers of naïve CD4 T-cell activation and proliferation compared to their CD123^+^ counterpart. pDCs expressed maturation makers at much lower levels and were poor at T-cell stimulation. In the absence of pathogens, naive T cells cultured with ASDCs produced higher levels of IL-4, IL-5, IL-13, IL-9, IL-17, IL-22 and IL-10 than those cultured with pDCs, indicating a polarisation towards Th2, Th9, Th17, Th22, and Treg, which have broad roles in humoral, bacterial and auto-immunity, and less towards Th1, responsible for antiviral responses [110]. This pattern of polarisation was reported to be linked to the ASDCs’ high expression of CD5. CD5 is required for DC2 to stimulate naïve T-cell proliferation and priming towards Th2, Th17, Th22 and Treg cells, while monocyte-like DCs, which express lower CD5, polarise towards Th1 cells [111]. Warner van Dijk et al. showed that HIV infection downregulated CD5 expression [87]. However, whether this downregulation favours the induction of the antiviral Th1 subset was not investigated due to low isolated ASDC numbers. Unlike pDCs, HIV-exposed ASDCs do not produce IFN-I [87]. This correlates with their lack of TLR7 and IRF7 expression [44,73]. Furthermore, their Siglec-1 expression may inhibit IFN production by inducing TBK1 degradation, as reported for macrophages [112].

In the CSF of patients with inflammatory demyelinating diseases, ASDCs were observed to have poly-adhesion functions with the ability to stimulate both CD4 T cells and LAMP3^+^ DCs (mature/migratory DC) and bind various immune cells, including B and T cells [106]. The authors concluded that ASDCs significantly contribute to the pathogenesis of conditions like MS by triggering an inflammatory cascade of immune cells. The further investigation of ASDCs in the context of specific inflammatory diseases is needed to uncover condition-specific functionalities.

### 6.3. HIV Uptake, Infection and Transfer to T Cells by ASDCs

Both the CD11c^+^ and CD123^+^ ASDC subsets can transmit HIV to T cells via differing mechanisms. Ruffin et al. [64] and Warner van Dijk et al. [87] found that CD123^+^ ASDCs isolated from blood were productively infected with HIV and transferred the virus to CD4 T cells via second-phase transfer. This correlates with their higher expression of the HIV entry receptors CD4 and CCR5. Warner van Dijk and colleagues further showed in blood-derived ASDCs that the CD11c^+^ subset was most efficient at mediating first-phase HIV transfer, corresponding with their higher expression of lectin receptor MR [56], langerin [54] and DC-SIGN [113]. Mature DCs have a higher capacity to bind HIV compared to immature cells [114] but are less capable of supporting productive infection [37,53,64]. Since CD11c^+^ ASDCs exhibit a more mature phenotype, characterised by high CD86 and HLA-DR expression [87], this likely explains their enhanced efficiency in mediating first-phase viral transfer. Additionally, CD11c^+^ ASDCs expressed higher levels of the HIV restriction factor SAMHD1, which may further explain their lack of productive infection. Brouiller et al. [109] investigated the HIV infection of blood-derived ASDCs and observed high p24 levels within CD11c^+^ ASDCs, indicating efficient productive infection. However, Warner van Dijk et al. pointed out that this productive infection was mediated by Vpx, which neutralises SAMHD1. This supports their conclusion that SAMHD1 expression in CD11c^+^ ASDCs limits productive infection.

The HIV infection of skin and mucosal ASDCs has remained unexplored due to low cell numbers. However, a comparison of HIV-binding receptor expression between human-blood and anogenital-tissue ASDCs was performed and revealed both tissue- and blood-derived ASDCs express CD4 and Siglec-1 at similarly high levels [87]. DC-SIGN, MR and langerin were more highly expressed by tissue ASDCs compared to blood, and more so by CD11c^+^ ASDCs than CD123^+^ ASDCs, however it is widely recognised that blood and tissue DCs express different levels of the same markers [38]. This purely emphasises the need to investigate HIV interactions with ASDCs at the primary transmission sites and further understand their relative importance in HIV infection compared to other tissue DCs. If tissue-derived ASDCs show similar profiles to blood ASDCs in HIV uptake, susceptibility to HIV infection, and mode of HIV transfer to T cells, future strategies aiming at inhibiting HIV transmission, namely PrEP regimes, should consider the interactions of ASDCs with HIV to prevent its transfer to CD4 T cells.

## 7. DC3

### 7.1. Origins and Discovery

Most recently, DC3s were discovered using scRNA sequencing in human blood by Villani et al. in 2017 [44]. DC3s are best defined as CD1c^+^-, CD163^+^-, CD14^+/−^-,CD5^−^- and CD88^−^-expressing DCs and for many years were misidentified due to their co-expression of defining markers for DC2s and monocyte-derived cells [45,46,47,115]. They have now been further identified in bone marrow [47], body skin [47,116], spleens [47], bronchoalveolar lavage [108], cancerous tumours [117,118], the synovium, synovial fluid [119], kidneys [120], the cervix and the endometrium [121] (Table 2).

DC3s are a bona fide DC population and develop from a distinctly different lineage to DC1s and DC2s. Like monocytes, they arise from an IRF8^low^ GMPD progenitor pathway, which is separate to the IRF8^high^-dependent GMPD pathway of DC1s, DC2s and pDCs [47]. Furthermore, Bourdely and colleagues [46] showed that DC3 differentiation is only driven by a GM-CSF-dependent pathway as opposed to an FLT3L-dependent pathway, the latter of which supports the differentiation of cDP. However, several cancer-specific investigations observed an in vitro tumour microenvironment phenomenon where DC2s shifted towards a DC3 like-phenotype, acquiring CD14 and CD163 expression, in response to tumour-derived factors including IL-6 and M-CSF [118,122]. This same shift was not observed in monocytes. It is likely that DC3s are indeed a bona fide population but in specific tumour environments may be capable of arising from highly plastic DC2s. Whether this DC2-DC3 shift is possible in other specific inflammatory microenvironments remains to be investigated.

### 7.2. Inflammation and Immunological Functions

DC3s have been specifically implicated in a number of inflammatory disease settings, with many studies highlighting a strong association between CD14 expression and inflammation. CD14-expressing DC3s are increased in the peripheral blood of patients living with chronic autoimmune diseases and systemic infections, including lupus erythematosus [45], melanoma [123] and severe COVID-19 [124]. Although the precise immune roles of inflammatory DC3s in these conditions remain unclear, blood DC3s have been shown to stimulate naïve CD4 and CD8 T cells [44,45,46], secrete pro-inflammatory cytokines such as IL-8, IL-23, TNFα and IL-1β [46,47] and induce Th1, Th2 and Th17 polarisation [45,46]. These immune capabilities are similar to DC2s. CD14^+^ DC3s were also more abundant in inflamed bronchoalveolar lavage when compared to a non-inflamed control [108]. Upon entering the inflammatory airspace, DC3s were functionally altered and capable of inducing IFNγ and IL-17 secretion from CD4 T cells, which stimulates Th1 and Th17 polarisation, respectively, supporting the previous findings of blood-derived DC3s.

In inflamed tissues, DC3s are significantly enriched and frequently associated with poor patient outcomes and treatment unresponsiveness across specific disease contexts [118,120,123]. Poor renal prognosis in lupus nephritis patients was directly linked to increased DC3 numbers in the kidney, driven by their potent capacity to prime Th1 and Th17 responses, causing tissue damage and T-cell trafficking [120]. Within malignant tumour microenvironments [117] and the synovium of osteoarthritis patients [119], DC3s also demonstrated strong Th1 priming capacity through high production levels of IL-12 and IL-18, and TNFα. Furthermore, synovial DC3s were efficient at activating CD8 T cells. However, in certain malignancies, DC3s exhibited impaired T-cell activation compared to DC2s, likely influenced by the tumour-specific secretome, which disrupts DC function [118]. Inflammatory DC3s identified in lesional body skin of psoriasis and atopic dermatitis patients expressed IL-6 and IL-23, indicating their potential role in Th17 cell differentiation [116], although it should be noted that the protein data from this investigation may be contaminated with B cells. Nakamizo et al. identified DC3s as CD14^+^ CD1c^+^ cells; however, CD14, CD3 (T cells) and CD19 (B cells) were all detected with the same fluorophore, which is confounding as a small population of memory B cells have also been reported to express CD1c in blood and lymphoid organs [125,126,127,128,129].

Based on the above, it is likely that many DC3 populations have been misidentified. In fact, a recent review by Odell found that out of 13 scRNA seq investigations published over the last 5 years investigating inflammatory scleroderma of the skin and lung and liver cirrhosis, only one accurately identified the DC3 population, and the remaining mislabelled DC3s as macrophages or monocytes. Upon re-analysis, Odell reclassified the misidentified populations as DC3s and demonstrated that DC3s were enriched in scleroderma and cirrhosis conditions, and their infiltration often correlated with disease severity. Without this re-analysis, the presence of these inflammatory DC3s would have been missed.

It is evident that DC3s are emerging as a central inflammatory DC with diverse immune functions. Investigations have demonstrated, both in human blood and tissue, that DC3s can stimulate naïve CD4 and CD8 T cells and prime Th1 and Th17 responses, and that they are potent pro-inflammatory cytokine secretors. Research must further elucidate the presence and function of DC3s in human skin and mucosae and their relation to other inflammatory pathologies.

### 7.3. HIV Interactions

Investigations examining the interactions of DC3s with HIV are limited, and little information exists on their expression of HIV-binding receptors. A very recent study by Parthasarathy and colleagues [121] investigated the changes in the gene and protein signatures of CD14^+^ DCs, as a combined population of DC3s and CD14^+^ MDDCs, before and after HIV infection in cervical and endometrial tissue. Upon HIV exposure, they observed an upregulation of genes associated with the initiation of inflammation and antiviral roles; however, they were unable to determine whether these genes were solely attributable to DC3s. Nonetheless, they demonstrated that DC3s had a high protein expression of the HIV entry receptors CD4, CCR5 and CXCR4, indicating the capacity for second-phase transfer and productive infection. A high expression of the HIV co-receptor CXCR4 has also been shown on DC3s in blood [44] and malignant tumours [117]. Villani et al. [44] showed at the RNA level that blood DC3s express Siglec-1 but not as highly as ASDCs. Dutertre et al. [45] confirmed this expression of Siglec-1, further evidencing this increased expression specifically on inflammatory DC3s.

With the striking similarities between the immune functions of DC3 and DC2, it is tempting to speculate over the potential involvement of DC3 in HIV transmission, given DC2 can efficiently transfer virus to CD4 T cells [38,43]. However, this remains a critical gap in the HIV-DC literature.

## 8. Other Inflammatory Tissue Dendritic Cells

### 8.1. Monocyte-Derived Dendritic Cell

Traditionally CD14 expressing cells were described as either autofluorescent macrophages or non-autofluorescent CD14^+^ DCs [130]. McGovern et al. [49] redefined the latter as a transient population of monocyte-derived macrophages (MDMs), which originated from blood-derived CD14^+^ monocytes. In 2021, Rhodes et al. [38] demonstrated that there are two populations of CD14^+^-expressing MNPs: a CD14^+^ CD1c^−^ CD11c^−^ MDM population that was non-migratory and transcriptionally and morphologically similar to macrophages, and a CD14^+^ CD1c^+^ CD11c^+^ MDDC population that could spontaneously migrate out of tissue and was transcriptionally and morphologically similar to DCs. This description of a CD14^+^ MDDC population aligned with previous reports of a CD14^+^ monocyte-derived cell transcriptionally aligning with DCs rather than blood monocytes [131,132,133,134,135] and possessing dendritic morphology [135,136].

MDDCs are produced at tissue sites during inflammation from infiltrating monocytes or in vivo in the presence of GM-CSF [137,138,139] and have been identified in a range of inflammatory conditions, including the pleural effusions of tuberculosis patients [140], urine of kidney transplant recipients with infection [141], intestinal lamina propria of Crohn’s patients [142], psoriatic skin [138] and inflamed ascites fluid, synovial fluid and spleens [115,136]. In human cervical tissue, inflammation was correlated with an increased CD14^+^ MDDC infiltrate [143]. Segura et al. [136] characterised a specifically inflammatory DC, termed infDC, distinguished by their expression of CD1c and CD14. These infDCs had a DC-like morphology, were transcriptionally distinct to other known DCs, were monocyte-derived and potently induced the Th17 differentiation of CD4^+^ memory T cells. Other studies have confirmed this strong Th17 response by MDDCs through IL-23 secretion [140], as well as Th1 polarisation through IL-12 secretion [132,133,144].

Like DC2s and DC3s, MDDCs are potent APCs, capable of antigen presentation and migration from tissues to lymph nodes, making them key players in HIV transmission [38,143]. Several studies observed a CD14^+^ CD11c^hi^ MDDC in human cervical tissue that was capable of capturing HIV using lectin receptors without the integration or replication of the virus. Rodriguez-Garcia et al. [145] was the first to demonstrate that tissue-derived MDDCs could capture HIV and showed that this occurred with or without the expression of DC-SIGN, implicating the presence of another lectin receptor mediating uptake. However, it is now known that the CD14^+^ CD11c^+^ DC-SIGN^+^ cells were in fact MDMs, as MDDCs are DC-SIGN^−^ [38]. Trifonova et al. further found MDDCs to be the most efficient mediator of first-phase transfer, particularly when compared to tissue-resident macrophages. MDDCs expressing Siglec-1 were found to efficiently transfer HIV to CD4 T cells by Perez-Zsolt and colleagues [143], which was partially blocked by Siglec-1 antibodies. These Siglec-1 expressing MDDCs were more abundant in inflammation. The study showed that IFN-α stimulated the production of Siglec-1, thereby promoting HIV uptake and first-phase transfer in inflamed environments. The authors hypothesised that pDC recruitment to inflammation sites, triggering an antiviral IFN-I response, drives the upregulation of Siglec-1. In support of the previous cervical studies, Rhodes et al. showed in human skin and anogenital tissues, MDDCs could preferentially take up, become productively infected by and transfer HIV to CD4 T cells. They further demonstrated that blocking up to 40% of HIV infection in in vivo MDDCs was achieved using a siglec-1 antibody. The investigation concluded that these MDDCs did not express DC-SIGN or Langerin, indicating perhaps an unknown lectin receptor was responsible for mediating HIV transfer [146]. Ex vivo tissue-derived MDDCs are also highly efficient at HIV transfer to CD4 T cells via the CD4/CCR5 mediated second-phase transfer pathway [38].

### 8.2. Epidermal CD11c^+^ Dendritic Cell

In addition to LCs, the stratified squamous epithelium of human skin and type II mucosal tissues contains other DCs. Known for being CD11c^+^ Langerin^lo^ and lacking Birbeck granules, this DC has been described in the literature under several different aliases including inflammatory dendritic epithelial cell (IDEC) [147], vaginal dendritic epithelial cell (VDEC) [148], Epidermal CD11c^+^ DC, and LC2, all probably representing independent identifications of the same cell population [149]. In the late 1990’s Wollenberg and colleagues [147] were the first to describe an epidermal DC present in inflammatory skin conditions such as atopic eczema and psoriasis. Critically, these cells were not LCs as they differed in expression profile and lacked Birbeck granules, however the functions of IDECs remain largely unexplored and unknown. It was proposed that in vitro generated IDECs are capable of initiating Th1 cell differentiation [150], however this has yet to be shown in vivo. Whilst IDECs were never explicitly investigated in relation to HIV, they express high levels of MR. Pena Cruz et al. [148] identified a distinct DC subset within healthy human vaginal epidermis termed VDEC which could become productively infected by HIV using the CD4/CCR5 entry pathway. In 2019 Bertram et al. identified an epidermal HIV transmitting DC, particularly identifiable by their high expression of CD11c and MR and transcriptionally indistinguishable from dermal DC2s. They demonstrated in abdominal tissue that epidermal CD11c^+^ DCs were present in lower proportions than LCs while in genital skin they were present in equal numbers. However, they predominated in type I and II mucosae. Compared to LCs, epidermal CD11c^+^ DCs were more efficient at HIV uptake and more susceptible to infection and correspondingly efficient at mediating both first-phase and second-phase transfer to CD4 T cells. The most recent independent identification of an epidermal DC population was described by Liu et al. as LC2s, sharing the same CD11c^hi^ Langerin^lo^ profile as all previously mentioned studies and expression of the transcriptional factor IRF4, which is highly associated with DC2s. LC2s were more abundant in foreskin epidermis and enriched in psoriatic skin inflammation, although they were not investigated in the context of HIV. In 2023 Bertram et al. used high parameter flow cytometry to demonstrate that IDEC, VDEC, epidermal CD11c^+^ DC and LC2 as originally identified, all correlated to the same cell type [149]. Although Pena-Cruz et al. and Bertram et al. have both showed these are key HIV target cells, their specific immune functions and interactions with HIV in an inflamed environment remain poorly understood.

## 9. Concluding Remarks

HIV remains a significant global health burden, and as of 2023 there were almost 40 million people living with HIV and 1.3 million new infections [151]. Notably, HIV transmission is strongly linked to inflamed anogenital mucosa [6,7,8], a consequence of pre-existing STI’s or microbiota dysbiosis [11,12,13,14,15,16], which frequently presents asymptomatically and hence remains undiagnosed [21,22]. Despite significant progress over the past 40 years, with reduced incidence owing to revolutionary prophylactic measures and treatments, recent studies have shown that PrEP can be ineffective in the presence of anogenital inflammation [17,18]. Critically, a vaccine or cure for HIV remains elusive.

The early events of HIV transmission in anogenital tissues, particularly in an inflammatory setting, remains relatively unknown. DCs are the first cells to interact with HIV, hence understanding their ontology, immune functions, expression profiles and interactions with HIV forms a crucial step in developing new treatments. However, investigations of in vivo inflammation are challenging to navigate as complex inflammatory signals create a dynamic and unpredictable cellular environment. In inflamed anogenital tissues, increased susceptibility to HIV infection is thought to result from compromised epithelial barrier integrity, which exposes a range of DCs in the underlying tissue [32,33,34]; changes to resident DCs, which heighten their vulnerability to infection [35,36]; and an influx of inflammation-specific HIV target cells [39,40]. The infiltrating inflammatory pDCs act as a ‘double-edged’ sword in their potential as a therapeutic target. pDCs’ potent antiviral IFN-I production [86] and pro-inflammatory CCL3-5 secretion, which blocks CCR5 [98], works as a vital defensive mechanism to prevent viral transmission. Designing therapeutic interventions to enhance the immunological functions of pDCs may enhance the body’s natural anti-HIV immune response [152]. However, simultaneously, this rapid IFN production induces the expression of Siglec-1 and drives the HIV binding and infection of many other DCs, amongst other limitations. Hence, any treatment targeting pDCs needs careful and innovative design.

ASDCs, CD14^+^ MDDCs and epidermal CD11c^+^ DCs are all capable of binding HIV and transmitting the virus to CD4 T cells in first- or second-phase transfer. Thus far, these inflammatory DCs have only been investigated in isolation. Future research should investigate the HIV infectivity of these DCs by comparing them to each other and to other resident DCs, such as DC2s, to determine a preferential HIV target cell. The ability of DC3s to interact with HIV remains a critical and unresolved question. A comprehensive exploration and profiling of inflammatory DCs, especially in human anogenital tissues, is essential for developing modified PrEP regimens to block DC infection via an inflamed mucosa. Phenotypic characterisation will help identify the specific receptors involved in binding and mediating HIV transmission, exposing the key receptors to target in prophylactic interventions. Closing this knowledge gap hinges on the difficulty of obtaining human anogenital tissue, particularly inflamed samples, and extracting a sufficient yield of viable cells. However, with the need for more effective HIV prevention strategies, this challenge is a necessary scientific pursuit.

## Figures and Tables

**Figure 1 viruses-17-00105-f001:**
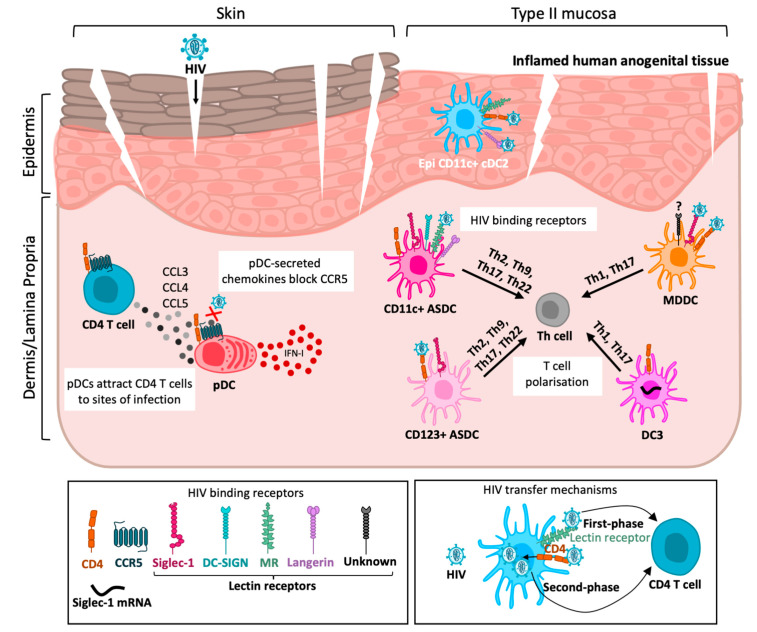
Inflammatory DC interactions with HIV in anogenital tissue. HIV enters inflamed anogenital tissue through breaches in the epithelium. Upon HIV exposure, plasmacytoid dendritic cells (pDCs) secrete IFN-I to induce an antiviral immune response and chemokines CCL3-5, which recruit CD4 T cells to infection sites and block the binding of HIV to the CCR5 co-receptor. Both Al^+^ siglec-6^+^ dendritic cell (ASDC) subsets are capable of polarising T cells into Th2, Th9, Th17 or Th22. CD11c^+^ ASDCs tended to be more efficient at first-phase transfer via HIV bending to lectin receptors (MR, langerin, DC-SIGN and Siglec-1), whilst CD123^+^ ASDCs tend to be more efficient at second-phase transfer via the entry receptors CD4/CCR5 and productive infection. DC3s can induce Th1 and Th17 polarisation. It has not yet been determined if they are capable of HIV binding and transfer; however, they express the HIV entry receptors CD4/CCR5 and Siglec-1 mRNA. Monocyte-derived dendritic cells (MDDCs) can also induce Th1 and Th17 polarisation. They can mediate first-phase transfer via Siglec-1 and likely another unidentified lectin receptor. MDDCs are also highly efficient at CD4/CCR5-mediated second-phase transfer. Epidermal CD11c^+^ DCs can efficiently bind and transmit HIV through both first- and second-phase transfer.

**Figure 2 viruses-17-00105-f002:**
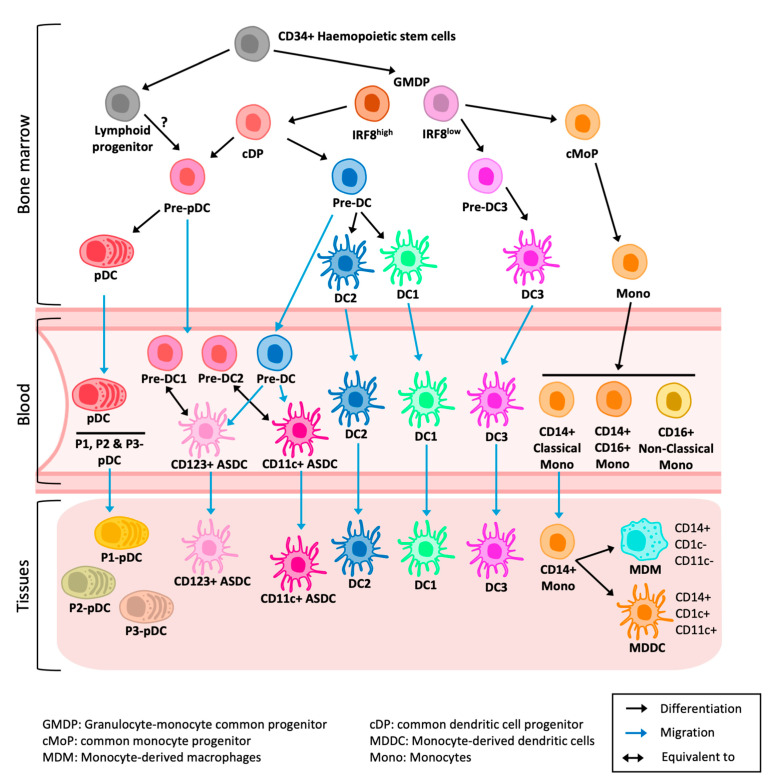
CD34^+^ hematopoietic stem cells produce granulocyte-monocyte common progenitor (GMDP) and lymphoid progenitors. Based on IRF8 expression, GMDP gives rise to the common DC progenitors (cDP), pre-DC3 and the common monocyte progenitor (cMoP). While cDP generate DCs (pDCs, ASDCs, DC1, DC2), pre-DC3 gives rise to DC3 and cMoP gives rise to monocytes. The developmental pathway of ASDCs from pre-pDCs or directly from pre-DCs remains to be fully characterised. PDCs and ASDCs migrate from blood to inflamed tissues. DC1, DC2 and DC3 are present in healthy and inflamed tissues but become more enriched in inflamed tissues. In inflammation, CD14 monocytes migrate from blood to tissues, and they differentiate into MDM and/or MDDCs.

**Figure 3 viruses-17-00105-f003:**
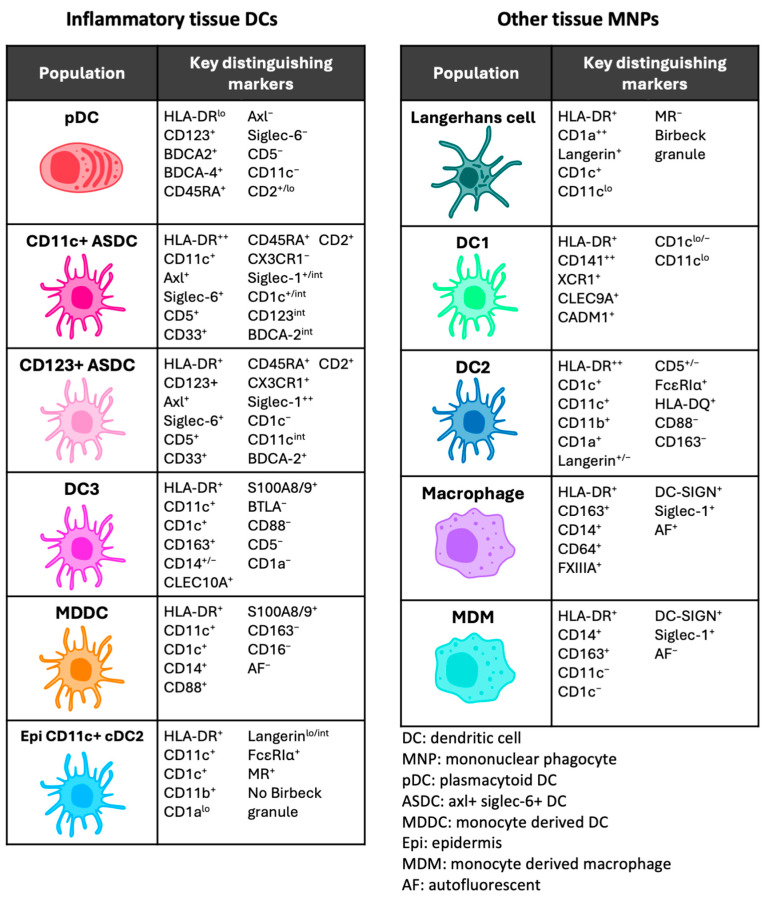
Human inflammatory and steady-state mononuclear phagocyte phenotypes.

**Table 1 viruses-17-00105-t001:** Summary of ASDCs in the current literature.

Citation	Source Tissue	Defining Markers	Separated into CD11c^+^ and CD123^+^ Subsets?	Immune Functions	HIV Interactions
Villani et al., 2017 [44]	Blood and tonsil	Axl^+^ Siglec-6^+^ CD5^+^ CD11c^+/−^ CD123^+/−^	Yes	Both subsets are potent stimulators of CD4 and CD8 T-cell proliferation	Not investigated
See et al., 2019 [89]	Blood and spleen	Siglec-6^+^ CD123^+^ CX3CR1^+^ CD45RA^+^ CD33^+^ CD5^+^ CD2^+^	No	Induced proliferation and polarisation of CD4 T cells	Not investigated
Alcántara-Hernández et al., 2017 [104]	Blood, tonsil and spleen	Axl^+^ CD123^+/int^ CD11c^+/−^ CD2^+^	Yes	ASDCs have a higher T-cell-stimulating capacity compared to pDCs	Not investigated
Warner van Dijk et al., 2024 [87]	Blood and anogenital tissue	Axl^+^ Siglec-6^+^ CD5^+^ CD11c^+/−^ CD123^+/−^ CX3CR1^+/−^	Yes	CD11c^+^ ASDCs are more potent inducers of CD4 T-cell activation and proliferation compared to CD123^+^ ASDCs. Both ASDC subsets polarise T cells towards Th2, Th9, Th17, Th22 and Treg	CD11c^+^ ASDCs are more efficient at first-phase transfer to CD4 T cells CD123^+^ ASDCs are more efficient at second-phase transfer to CD4 T cells
Kang et al., 2023 [106]	Cerebrospinal fluid (demyelinating diseases)	Axl^+^ Siglec-6^+^	No	Stimulate CD4 T cells and mature LAMP3^+^ DCs Bind B and T cells	Not investigated
Chen, 2020 [107]	Skin (blisters and wounds)	Axl^+^ Siglec-6^+^ BDCA-2+ CD123^int^	No	Identified as an early infiltrator in inflammation	Not investigated
Jardine, 2019 [108]	Bronchoalveolar lavage	Axl^+^ Siglec-6^+^	Yes	Too few cell numbers for functional investigation	Not investigated
Ruffin, 2019 [64]	Blood	Axl^+^ CD123^+^ CD45RA^+^ Siglec-1^+^	No	Not investigated	CD123^+^ ASDCs are productively infected with HIV and transfer the virus to CD4 T cells
Brouiller, 2023 [109]	Blood	Axl^+^	Yes/No (investigated as both combined and separate)	Not investigated	Productive HIV onfection of ASDCs was mediated by Vpx which neutralises SAMHD1, a restrictive factor that limits productive infection

**Table 2 viruses-17-00105-t002:** Summary of DC3s in the current literature.

Citation	Source Tissue	Defining Markers	Immune Functions	HIV Interactions
Villani et al., 2017 [44]	Blood and tonsil	CD1c^+^ CD163^+^	Stimulates naïve CD4^+^ and CD8^+^ T cells	High protein expression of HIV co-receptor CXCR4 RNA expression of HIV lectin receptor siglec-1
Dutertre et al., 2019 [45]	Blood (lupus erythematosus)	CD1c^+^ CD163^+^ CD5^−^ CD88^−^ CD14^+/−^	Stimulates naïve CD4^+^ T cells. Induces Th1, Th2 and Th17 polarisation	RNA expression of HIV lectin receptor siglec-1 which is increased on inflammatory DC3s
Bourdely et al., 2020 [46]	Blood and primary breast tumours	CD1c^+^ CD163^+^ CD88^−^ CD14^+/−^	Stimulates naïve CD4^+^ and CD8^+^ T cells Secretes pro-inflammatory cytokines IL-23 and TNFα	Not investigated
Cytlak et al., 2020 [47]	Blood, spleen, dermis	CD1c^+^ CD163^+^ CD14^+^ BTLA^−^	Secretes pro-inflammatory cytokines IL-8, TNFα and IL-1β	Not investigated
Nakamizo et al., 2021 [116]	Blood and body skin (psoriasis)	CD1c^+^ CD14^+^ CD88^−^	Co-expression of markers IL-6 and IL-23 indicates their potential role in Th17 cell differentiation	Not investigated
Jardine et al., 2019 [108]	Bronchoalveolar lavage	BTLA^-^ CD5^-^ CD163^+^ CD14^+^ S100A8/9^+^	Induces Th1 and Th17 polarisation	Not investigated
Chen et al., 2024 [120]	Kidney (lupus nephritis)	CD163^+^ CD1c^+^ CD88^-^	Induces Th1 and Th17 polarisation. High DC3 numbers associated with poor renal prognosis	Not investigated
Subtil et al., 2024 [118]	Primary malignant colorectal tumor and liver metastasis	CD14^+^ CD1c^+^ CD163^+^	Impaired T-cell-activating and -proliferating capacity compared to DC2s	Not investigated
Santegoets et al., 2020 [117]	Oropharyngeal squamous cell carcinoma tumor	CD1c^+^ CD163^+^ CD14^−^	Secretes cytokines IL-12 and IL-18 Primes Th1 polarisation	High protein expression of HIV co-receptor CXCR4
Qiu et al., 2022 [119]	Synovium and synovial fluid (osteoarthritis)	CD1c^+^ CD163^+^ CD88^−^	Secretes pro inflammatory cytokines TNFα, IL-23 and IL12p70 Primes CD8+ T cells	Not investigated
Parthasarathy et al., 2024 [121]	Cervix and endometrium	CD1c^+^ CD14^+^ (combined with MDDC)	Upregulation of genes associated with the initiation of inflammation and antiviral roles	High protein expression of HIV entry receptors CD4, CCR5 and CXCR4

## Data Availability

Not applicable.

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
