# Peer review of "Recent Advances in Our Understanding of Human Inflammatory Dendritic Cells in Human Immunodeficiency Virus Infection"

_viruses, 2025, doi:10.3390/v17010105_

Round 1
Reviewer 1 Report
Comments and Suggestions for Authors
The authors Warner Van DiJk et. al provide a very nicely prepared comprehensive review of the current state of our understanding of human DC biology, with a focus on inflammatory dendritic cell subsets in the setting of HIV infection. The manuscript was well organized and well written, and quite informative
I have a few very minor comments.
11. In figure 1, I suggest that the smaller boxed images showing the HIV binding receptors and the HIV transfer mechanisms be expanded in size so that the fonts can be large enough to read when the paper is printed out. Even with glasses on, I had trouble seeing the text unless I had it enlarged on the computer screen. I would have the fonts be no smaller than what is in the upper image.
22. On page 6 line 164, where the authors mention that “DC-SIGN is highly efficient at HIV binding and uptake…,” I feel that this statement should use a reference, such as that from Geijtenbeek et al, Cell 2000 (PMID 10721995)
33. On the same page, line 183, the references at the end of the paragraph should be combined, so instead of showing as (66) (67, 68) it should be (66-68).
44. Throughout the manuscript, the authors refer to people living with HIV or people with HIV as “HIV patients” as shown on lines 265 and 269. The authors should go through the manuscript to search where they refer to people with HIV as “patients” and re-phrase. This is a somewhat sensitive issue among the HIV community, I’ve listed the URL address for the UNAIDS TERMINOLOGY GUIDELINES for 2024 to refer to if needed.
https://www.unaids.org/en/resources/documents/2024/terminology_guidelines
55. In Tables 1 and 2, the title for the first column is listed as “Author”. I suggest changing that to “Citation” since it is really the entire citation and citation number, rather than just a first author.
Author Response
Reviewer 1:
The authors Warner Van DiJk et. al provide a very nicely prepared comprehensive review of the current state of our understanding of human DC biology, with a focus on inflammatory dendritic cell subsets in the setting of HIV infection. The manuscript was well organized and well written, and quite informative
I have a few very minor comments.
In figure 1, I suggest that the smaller boxed images showing the HIV binding receptors and the HIV transfer mechanisms be expanded in size so that the fonts can be large enough to read when the paper is printed out. Even with glasses on, I had trouble seeing the text unless I had it enlarged on the computer screen. I would have the fonts be no smaller than what is in the upper image. The fonts have been amended as requested in Figure 1.
On page 6 line 164, where the authors mention that “DC-SIGN is highly efficient at HIV binding and uptake…,” I feel that this statement should use a reference, such as that from Geijtenbeek et al, Cell 2000 (PMID 10721995). PMID 10721995 has been added as ref 58
On the same page, line 183, the references at the end of the paragraph should be combined, so instead of showing as (66) (67, 68) it should be (66-68). These references are now combined on page 6 as 69-71.
Throughout the manuscript, the authors refer to people living with HIV or people with HIV as “HIV patients” as shown on lines 265 and 269. The authors should go through the manuscript to search where they refer to people with HIV as “patients” and re-phrase. This is a somewhat sensitive issue among the HIV community, I’ve listed the URL address for the UNAIDS TERMINOLOGY GUIDELINES for 2024 to refer to if needed. https://www.unaids.org/en/resources/documents/2024/terminology_guidelines. The term “HIV patients” is now replaced with “people living with HIV (PLH)”
In Tables 1 and 2, the title for the first column is listed as “Author”. I suggest changing that to “Citation” since it is really the entire citation and citation number, rather than just a first author. “Author” has now been replaced by “Citation” in tables 1 and 2.
We have also amended in figure 3 cDC1 to DC1 and cDC2 to DC2 to match what is the manuscript.
Reviewer 2 Report
Comments and Suggestions for Authors
This review by Warner van Dijk and colleagues summarizes the current knowledge on HIV - dendritic cells (DCs) interaction with specific focus on DCs in inflamed tissues. These ”inflammatory” DCs are postulated to play important roles in virus dissemination, immune activation, and inflammation, therefore understanding the mechanisms of DC-HIV interaction is crucial to elucidate HIV pathogenesis. This review is very well written, thorough, and comprehensive. There are some minor comments for clarification as follows:
Most of the DCs subsets discussed here are DCs infiltrating into inflamed tissues. It would be helpful to briefly discuss what triggers inflammation at the site of infection (role of tissue resident DCs?).
It would be helpful to add a figure depicting DC (and macrophage) lineages and development. Please ignore this comment if it’s too much work.
Line 152. “within 12-24 hours the virus is no longer found within these VCCs” is somewhat inaccurate and should be rephrased. The cited paper (#36) and others showed the presence of HIV/SIV 24 hours post virus addition to DCs. Also, although it is clear that most of particles will be degraded within 24 hours, it should be noted that even a small amount of virus can potentially contribute to T cell infection since infection via a virological synapse is very efficient.
Line 167. Siglec-1 expression can be triggered by type III interferon as well (PMID: 38589249).
Line 568. “iltrating” > infiltrating?
Author Response
This review by Warner van Dijk and colleagues summarizes the current knowledge on HIV - dendritic cells (DCs) interaction with specific focus on DCs in inflamed tissues. These ”inflammatory” DCs are postulated to play important roles in virus dissemination, immune activation, and inflammation, therefore understanding the mechanisms of DC-HIV interaction is crucial to elucidate HIV pathogenesis. This review is very well written, thorough, and comprehensive. There are some minor comments for clarification as follows:
Most of the DCs subsets discussed here are DCs infiltrating into inflamed tissues. It would be helpful to briefly discuss what triggers inflammation at the site of infection (role of tissue resident DCs?). We have now added on Page 3 the following “Factors that promote inflammation at the site of infection were recently summarised by Caputo et al (ref) and included the ability of different anogenital mucosae (and the pH of the mucus) to act as a physical barrier to inflammatory agents and whose integrity can be compromised by infections, especially sexually transmitted diseases (including herpes simplex virus), hormonal contraceptives and alterations in the microbiome composition”. We have also added on page 4: “Very little work has been done in the genital tract to define dendritic cell responses to viruses or other microbial products. Therefore, future investigations into which dendritic cells are responding to sexually transmitted viruses such as HSV-2 or pathogenic species present in dysbiotic genital tissue, such as Gardnerella vaginalis, or Prevotella may identify therapeutic targets to interrupt the ongoing recruitment of immune cells in response to pathogenic microbiomes”
It would be helpful to add a figure depicting DC (and macrophage) lineages and development. Please ignore this comment if it’s too much work. This is now included as figure 2 and with a sentence on page 5 “Dendritic cells and macrophages development are summarised in Figure 2”. To make the text more clear and align it with what is shown in this figure, we have also amended page 9 by replacing the sentence reading “ASDCs are also known as pre-DCs as they can be induced to differentiate into DC1 (CD123+) and DC2 (CD11c+) ref ” with “The CD123+ and CD11c+ ASDC populations were identified by Villani et al.(Ref) expressing AXL, Siglec-6, CD2, CX3CR1, CD33, and CD5, similar to the phenotype described for pre-DC1 and pre-DC2 (CD33+ CD45RA+ CD123+ CX3CR1+ CD2+ CD5+ Siglec-6+) identified by See et al. (Ref). Therefore, CD123+CD11c- ASDCs are the pre-DC1 while CD123- CD11c+ ASDCs are the pre-DC2 with CD11c+ ASDCs being a different population than DC2 (Ref). ASDCs infiltrate peripheral tissues during inflammation while cDC2 are present in both healthy (Ref) and inflamed tissues (Ref).
Line 152. “within 12-24 hours the virus is no longer found within these VCCs” is somewhat inaccurate and should be rephrased. The cited paper (#36) and others showed the presence of HIV/SIV 24 hours post virus addition to DCs. Also, although it is clear that most of particles will be degraded within 24 hours, it should be noted that even a small amount of virus can potentially contribute to T cell infection since infection via a virological synapse is very efficient. “within 12-24 hours the virus is no longer found within these VCCs” has been amended to read “within 12-24 hours the majority of the virus is degraded”
Line 167. Siglec-1 expression can be triggered by type III interferon as well (PMID: 38589249). We have now added on line 167 “Recently it was induced at the gene levels in macrophages by type III interferon (l3) although this remain to be validated at the at the protein level” with PMID 38589249 being ref 60.
Line 568. “iltrating” > infiltrating? Amended to fix the typo